

**Stepwise drying of Lake Turkana at the end of the African Humid Period:**
**an example of forced regression modulated by solar activity?**
**Alexis Nutz[1], Mathieu Schuster[1]**
*Institut de Physique du Globe de Strasbourg (IPGS), UMR 7516, Centre National de la*
*Recherche Scientifique, Université de Strasbourg, École et Observatoire des Sciences de la*
*Terre, 1 rue Blessig, 67084 Strasbourg, France*
*Corresponding author (nutz@unistra.fr)*
**Running head:** Lake Turkana drying at the end of the AHP
**Keywords:** East African Rift System; Solar activity; Turkwel delta; Lake level; Holocene
**Abstract**
Although timing of the termination of the African Humid Period (AHP) is relatively well-
established now, modes and controlling factors are still being determined. Here, through a
geomorphological approach, we characterize the evolution of the final regression of Lake
Turkana at the end of the African Humid Period. We show that lake level fall during this
period was not constant, yet rather stepwise consisted of five periods marked by rapid rates of
lake level fall separated by periods of lower rates of lake level fall. Even the overall regressive
trend is associated with regional decreased precipitations due to reduced insolation controlled
by orbital precession, we discuss the origin of the five periods of accelerated rates of lake
level fall. Finally, we propose that accelerations are associated with periods marked by solar
activity minima that locally resulted in the repeated westward displacement of the Congo Air
Boundary (CAB), thereby reducing rainfall across the Lake Turkana basin.



## 1. Introduction

The African Humid Period (AHP), *c.* 14.8 to 5.5 ka BP, is a major climate period that was

paced by orbital parameters (i.e. precession) (deMenocal et al., 2000; deMenocal and Tierney,

2012; Bard, 2013; Shanahan et al., 2015) and that markedly impacted environment,

ecosystems, and human occupation of Africa over several millennia (Bard, 2013). An increase

in rainfall during this climate period led to the rise and highstand of numerous African lakes

(Street and Grove, 1976; Tierney et al., 2011). The end of the AHP was characterized by the

establishment of more arid conditions, leading to dramatic lake level falls (Street-Perrott and

Roberts, 1983; Kutzbach and Street-Perrott, 1985). This aridification forced Neolithic

populations to adapt to more limited resources (Kuper and Kröpelin, 2006) and represents a

recent example of major climate change. Depending on the location, the AHP termination

occurred at variable time-scales (Shanahan et al., 2015), being either abrupt (deMenocal et al.,

2000) or gradual (Kröpelin et al., 2008), thereby highlighting the complex interactions among

the variable responses to dominant forcings and multiple components of the local

environment (e.g., deMenocal, 2000; Renssen et al., 2006; Liu et al., 2007; Tierney and

deMenocal, 2013; Shanahan et al., 2015). However, drying trends remains poorly-constrained

and as a consequence the final regressions of African lakes are presented at relative constant

rate of lake level fall. In this study, we investigate the drying trend of Lake Turkana and

evidence for the first time that the final regression was not continuous through time. Thus,

understanding the mode of African lake regressions appears as particularly relevant in the

context of projecting future global climate change impacts on the African continent (e.g.,

Patricola and Hook, 2011), especially in term of evolution of water resources from large

lakes.

Lake Turkana is one of the great lakes of the East African Rift. It is considered as a

Wind-driven Waterbody (Nutz et al., in press) that developed abundant and well-developed





wave-dominated coastal features all along its shoreline. Those coastal features provide
valuable sediment archives that participated to the understanding of the evolution of the Lake
Turkana level during the AHP (Garcin et al., 2012, Forman et al., 2014; Bloszies et al., 2015).
However, the detailed and continuous evolution of lake level during the final forced
regression marking the end of the AHP has not been already documented. Here, the delta
complex of the Turkwel River (Fig. 1) that developed during the final forced regression of
Lake Turkana is examined using trajectory analysis (Helland-Hansen and Hampson, 2009).
Finally, we interpret variations in the rate of lake level fall as markers reflecting variable rate
of decrease in precipitation during the crucial period corresponding to the terminal phase of
the AHP. Subsequently, we discuss potential forcings responsible for the regressive pattern of
Lake Turkana with a primary focus on the role of the Sun and short-term variability of
insolation.

## 2. Materials and methods

The data set is comprised of satellite imagery and a digital elevation model (DEM). A
recently obtained SRTM1 (Shuttle Radar Topography Mission (Rabus et al., 2003)) is
available for the entire Lake Turkana area. This DEM is produced by radar interferometry
with a one arcsec (approximately 30 m) horizontal grid spacing and multi-metre resolution in
the vertical dimension. In addition, high-resolution (<1 m) PLEIADES and (5 m) SPOT 5
images (©CNES 2012, Airbus DS/ SPOT Image) were used to focus on selected areas. This
data set was processed using GIS software (Global Mapper 15 software; Blue Marble
Geographics, Hallowell, ME, USA) to provide a high-resolution 3D view of the
geomorphological features. Topographic profiles, elevation differences, and slope values used
for the trajectory analyses were obtained using Global Mapper 15 software.

## 3. Chronological framework



Humid conditions related to the AHP broadly prevailed over Africa from 14.8 to 5.5 ka BP
(deMenocal et al., 2000; Shanahan et al., 2015). Several lake level curves associated with
Lake Turkana evolution provide records of the regional moisture history over the Holocene
(Garcin et al., 2012; Forman et al., 2014; Bloszies et al., 2015). Based on surveys of raised
Holocene beach ridges coupled with dated archeological sites, these studies provide a
relatively robust chronological framework for the final regression at the end of the AHP.
Garcin et al. (2012) initially estimated the onset of the final lake level fall in Lake Turkana at
*c*. 5.27 ± 0.36 ka. Subsequently, Forman et al. (2014) refined the age of this final regression
proposing that it occurred between 5.5/5.0 to 4.6 ka BP associated with a lake level change
from 440 to 380 m asl. Finally, Bloszies et al. (2015) proposed an onset of the final regression
of the AHP starting at 5.18 ± 0.12 ka BP (dating of a shell at 90 m above the modern Lake
Turkana) and finishing at 4.6 ± 0.3 ka BP (age reused from Forman et al., 2014) associated
with a lake level grading from 450 to 375 m asl. As such, based on the most recent available
age-model of Bloszies et al. (2015), the final regression of Lake Turkana at the end of the
AHP would, at the longest, span a period from 5.3 to 4.3 ka BP. At a minimum, the final
regression would have occurred between 5.06 and 4.9 ka BP. This implies a duration ranging
between 160 to 1000 years, with a mean duration of 580 years for water level to decrease
from the Holocene highstand (450 m asl) to the lowstand (375 m asl). Because the
investigated portion of the Turkwel delta is located between 450 and 375 m asl, ages of the
landforms are considered to have developed between 5.18 ± 0.12 and 4.6 ± 0.3 ka BP.
**4. Geomorphological analysis**
The Turkwel delta complex is 35 km long, forming one of the major deltaic systems that
fringed Lake Turkana during the Holocene (Fig. 1). It was developed as the shoreline
migrated basinward, lowering from 450 to 360 m asl (Fig. 2). From west to east, five distinct
progradational stages were identified (Fig. 2d). The first progradational stage forms a lobe



protruding out from the mean north–south paleoshoreline, well defined by the 450 m asl
elevation shoreline (red line in Fig. 2d). According to regional age models (Garcin *et al.*,
2012; Forman et al., 2014; Bloszies et al., 2015), this first progradational stage marks the last
Holocene highstand before the end of the AHP. Moving eastward, each of the three
topographic profiles cross-cutting the Turkwel delta complex (Fig. 3) shows four slightly
inclined plateaus interrupting at *c.* 445, 425, 410, 400 and 390 m asl, respectively, separated
by five abrupt 5-to 15-m-high steps (Fig. 4). Each plateau defines a different progradational
stage. The plateaus are 3- to 5-km-wide, and correspond to successively abandoned delta
plains (Fig. 2d). To the north, these plateaus systematically end with paleo-spits that
document ancient, northward-flowing alongshore currents. The resulting landform reveals the
Turkwel delta complex as composed of successive asymmetric wave-dominated deltas
(Bhattacharya and Giosan, 2003; Anthony 2015) during most of its evolution, except in the
early period associated with the AHP highstand. None of the plateaus exhibit any evidence of
significant erosion that would indicate reworking of the landforms subsequent to their
deposition, except for the fluvial incision of the Turkwel River that progressively adjusted to
the base level fall. This supports the Turkwel delta complex as a primary depositional
landform corresponding to a continuous, comprehensive record of lake level evolution.
Trajectory analysis, performed for the three transects that cross-cut the Turkwel delta complex
along its progradation axis (Fig. 3), reveals that the plateaus are continuous, having slightly
descending regressive trajectories (slope gradient: >0° to 0.4°). The five abrupt steps that
separate plateaus have much higher slope gradients (1° to 3.8°), and are also defined as
descending regressive trajectories. Trajectories reflect a general lake level fall that meets the
definition of a forced regression (Posamentier *et al.*, 1992). Nevertheless, the five abrupt steps
reflect recurrent, short-lived increases in the rate of lake level fall. This evidences a stepwise
forced regression at the end of the AHP. In order to confirm this interpretation, we



investigated another portion of the Lake Turkana paleoshoreline. In the eastern Omo River
valley (Fig. 1), topographic profiles along two fossil spits are presented (Fig. 5). The two spit
systems show successive plateaus at elevations (*c.* 445, 425, 410 and 400 m asl) similar to
those observed in the Turkwel delta complex (Fig. 3). Finally, these additional observations
support the evolution of lake level as deduced from the Turkwel delta complex and the overall
trend of the three transects in the Turkwel delta as well as transects in the fossil spits of the
eastern Omo River valley lend support to the idea of a stepwise final, forced regression of
Lake Turkana at the end of the AHP.
**5. Discussion**
**5.1. Origin of Lake Turkana lake level evolution**
Lake level fluctuations may result from changes in the quantity of water supply to a lake,
from altered evapotranspiration rates within the catchment area, or from modifications in
basin physiography.  These changes may originate from a number of potential external
forcing processes, among which the most commonly considered are tectonism and climate.
Tectonism may be ruled out as the origin of any physiographic modification of the Lake
Turkana basin that would have caused abrupt falls in lake level at such time-scale. Vertical
crustal movements occur over much longer time periods than that of the AHP termination and
the rate of subsidence in the basin is too low (i.e. 0.4 m·ka$^{-1}$ at the Eliye Spring well site
(Morley *et al.*, 1999)), to explain several lake level falls of >5 m each in maximum 1000
years. Moreover, vertical displacements at this scale would require earthquakes having a
magnitude >9 (Pavlides and Caputo, 2004). Earthquakes of this magnitude are unknown in
the area and are not compatible with rift systems. Finally, volcanism event is known to have
occurred (Karson and Curtis, 1994) during the Late Quaternary even the age is not very-well
constrained. However, repeated pulsed of accelerated subsidence related to successive
emptying of magma chamber is prevented by the insufficient amount of magma observed in



the basin. Indeed, no regional magmatic effusion that would have caused sudden subsidence
is observable. Magmatism rather corresponds to punctual effusion forming the north, central,
and south islands. As such, the abrupt nature of the accelerated lake level falls can be
attributed only with difficulty to tectonics and magmatism leaving climate variability as the
most likely forcing mechanism.
During the Holocene, the overall climate pattern in East Africa was governed by insolation
changes related to changes in precessional orbital parameters of the Earth (Barker et al.,
2004). Links between insolation and hydrology are now well established for this region, in
particular monsoonal rainfall intensity that is strongly correlated with summer insolation
(deMenocal et al., 2000; Shanahan et al., 2015). In the early Holocene, an increase in summer
insolation due to changing orbital parameters produced wetter conditions over much of the
African continent leading to the establishment of the AHP. Subsequently, the overall
contraction of lakes at the end of the AHP is generally attributed to decreased precipitation
related to a reduction of summer insolation (deMenocal et al., 2000; Shanahan et al., 2015)
controlled by orbital parameters (i.e. half precessional forcing; deMenocal and Tierney, 2012;
Bard, 2013). Therefore, changes in insolation imply additional modifications in rainfall
amounts through the strengthening or weakening of local climate processes. In the Lake
Turkana area, Junginger et al. (2014) suggest that the increase of precipitation during the AHP
is mainly a result of a north-eastward shift of the Congo Air Boundary (CAB).  The CAB is a
north-east to south-west oriented convergence zone presently located west of the Lake
Turkana area. This convergence zone shifts eastward during higher insolation periods in
response to an enhanced atmospheric pressure gradient between India and East Africa during
northern hemisphere insolation maxima (Junginger and Trauth, 2013; Junginger et al., 2014).
When the CAB moves eastward over the Turkana area, precipitation is expected to increase
significantly. Finally, the five abrupt accelerations in lake level fall require short-term



accentuated decreases in precipitation. We propose that these five periods of significantly
reduced rainfall amounts are related to short-term decreases of insolation that repeatedly
moved the CAB position. At such time-scale, variations of solar activity appear as the most
likely acting parameter to explain variations in insolation. This potential origin needs to be
discussed.
**5.2. Linking solar activity and paleohydrology**
Links between short-term (decadal-scale) solar activity and climate change remains a point of
debate.  However, periodicities in solar activity such as the 11-year sunspot cycle, the
Gleissberg cycle (80—90 years) (Peristykh and Damon, 2003) or the de Vries cycle (~200
years) (Raspopov et al., 2008) have been identified in Holocene paleoenvironmental records
and suggests a possible forcing by solar activity on climate (Crowley, 2000; Bond et al., 2001;
Gray et al., 2013). Within some African lakes, several authors link more arid periods with
solar activity minima (Stager et al., 2002 and Junginger et al., 2014) and Lake Turkana is one
of them. These lakes are considered as *amplifier lakes* (Street-Perrott and Harrison, 1985) that
correspond to lakes for which relatively moderate changes in climate are amplified by the
specific morphology of rift. As an amplifier lake, Lake Turkana could be more sensitive to
precipitation changes from small variations in insolation as those generated by modifications
in solar activity.
Coupling the chronological framework proposed by Bloszies et al. (2015) with the
solar activity curve from Steinhilber et al. (2009), we observed in the Lake Turkana between
two and ten major solar activity minima during the minimum and maximum potential period
of regression, respectively (Fig.6). Considering a mean time of 580 years given by the age-
model during which the final regression occurred, five solar activity minima are observed.
The number of these minima interestingly matched with the number of abrupt lake level falls



suggesting a possible link between the short-term variability of solar activity and the lake
level changes in Lake Turkana at the end of the AHP. Because a mechanism must be given,
we propose that periods of solar activity maxima would be able to compensate for the
precession-induced reduction of insolation. The relatively limited reduction of insolation
would have led to a relatively stable position for the CAB over the Lake Turkana area and, in
turn, a reduced rate of lake level fall due to slowly decreased precipitation amounts. However,
when short-term solar activity minima are coupled with the precession-related insolation
decrease, the CAB would have migrated rapidly westward resulting in drastic reduction of
rainfall and as a consequence, a rapid fall in lake level. As such, alternations of solar activity
maxima and minima could explain the geomorphological pattern that revealed a long-term fall
in lake level interspersed by short-term accelerations in the rate of lake level fall during the
final forced regression at the end of the AHP.
**6. Conclusion**
Geomorphic analysis (i.e. trajectory analysis) revealed for the first time a stewise lake level
fall of Lake Turkana during the final forced regression of the lake at the end of the AHP. Five
rapid falls in lake level were identified, intercalated with periods of slower lake level fall. The
abrupt accelerations of lake level fall may be associated with insolation minima altering the
position of the CAB, responsible for regional precipitation pattern. Our interpretation suggests
that short-term variability of insolation, due to variability in solar activity, may have
influenced the hydroclimatic conditions in the Turkana area during the final forced regression
of the AHP. Next step would be to correlate each paleo plateaus to a specific solar maxima
and each step to a specific minima. Nevertheless, uncertainties of dating methods will allow
only with difficulty to provide enough precise ages for such features developed at the decadal
to centennial time-scale.




**Author contribution**

Alexis Nutz analyzed satellite images, co-writes the manuscript and participated to field work
Mathieu Schuster co-writes the manuscript and participated to the field work.

**Acknowledgements**

This works is a contribution of the Rift Lake Sedimentology project (RiLakS) funded by Total
Oil Company. Satellite images (SPOT and PLEIADES) were acquired thanks to the support
of CNES/ISIS program. Finally, we are grateful to Murray Hay (Maxafeau Editing Services)
for verification of the English text.

**The authors declare that they have no conflict of interest**

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

**Figure captions**
Figure 1. Location maps. (a) Lake Turkana basin in the East African Rift System (EARS). (b)
Digital elevation model (DEM) SRTM1 showing Lake Turkana and the two considered areas
(Turkwel delta and the east side of the Omo River valley). Dashed white line represents the
maximum Holocene lake level. All described geomorphological features are located between
the paleolake limit and the modern lakeshore.
Figure 2. Turkwel delta complex. (a) Raw digital elevation model SRTM1 of the Turkwel
delta. (b) Slope direction shading applied to the DEM SRTM1 of the Turkwel delta to
highlight the steps separating the different plateaus. Markers display the correspondence
between the DEM SRTM1 and the slope direction shading (see (a)). (c) SPOT5 satellite
image of the Turkwel delta. (d) Interpretative geomorphological map of the area showing five
successive delta plains in addition to the oldest plain associated with the late AHP highstand.
Figure 3. Geomorphological data for the Turkwel delta complex. (a) SRTM1 images were
processed to display a digital elevation model of the Turkwel delta complex. Locations of the
topographic transects are presented. (b) Topographic transects P1, P2, and P3. (c) Trajectory



analyses show that the overall forced regressive trend (>0° to 0.4°) is punctuated by four to
five steeper slopes (1° to 3.8°) revealing short-term increases in the rates of lake level fall.
Figure 4. Landforms from Turkwel delta. (a) Front view of a step grading downward to a
plateau. (b) Side view of the same step separating two plateaus.
Figure 5. Sandspit systems, outlined by dashed white lines, along the eastern Omo River
valley (location Fig.1b) from SRTM 1 (left side) and from PLEIADES images (right side).
The sandspits display plateaus having similar elevations as those of the Turkwel delta.
Figure 6. The red curve presents total solar irradiance (40-year moving average) relative to the
value of the PMOD composite during the solar cycle minimum of the year 1986 (1365.57
W.m²) (Steinhilber et al., 2009) for the period contemporaneous with AHP regression of Lake
Turkana. The shaded band represents 1σ uncertainty. The blue curve represents the
precessional curve covering the same time period
(http://www.imcce.fr/Equipes/ASD/insola/earth/online/). Grey stripes highlight solar activity
minima.








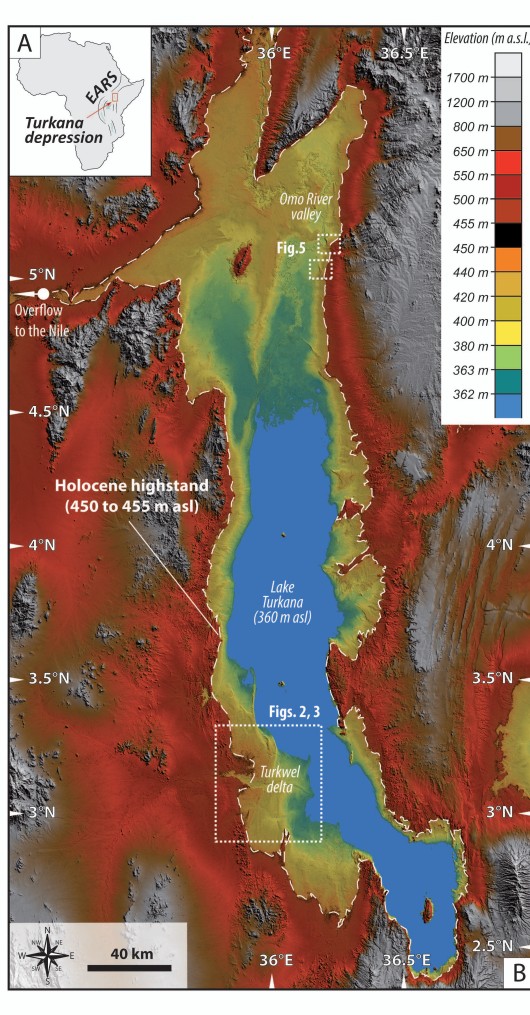





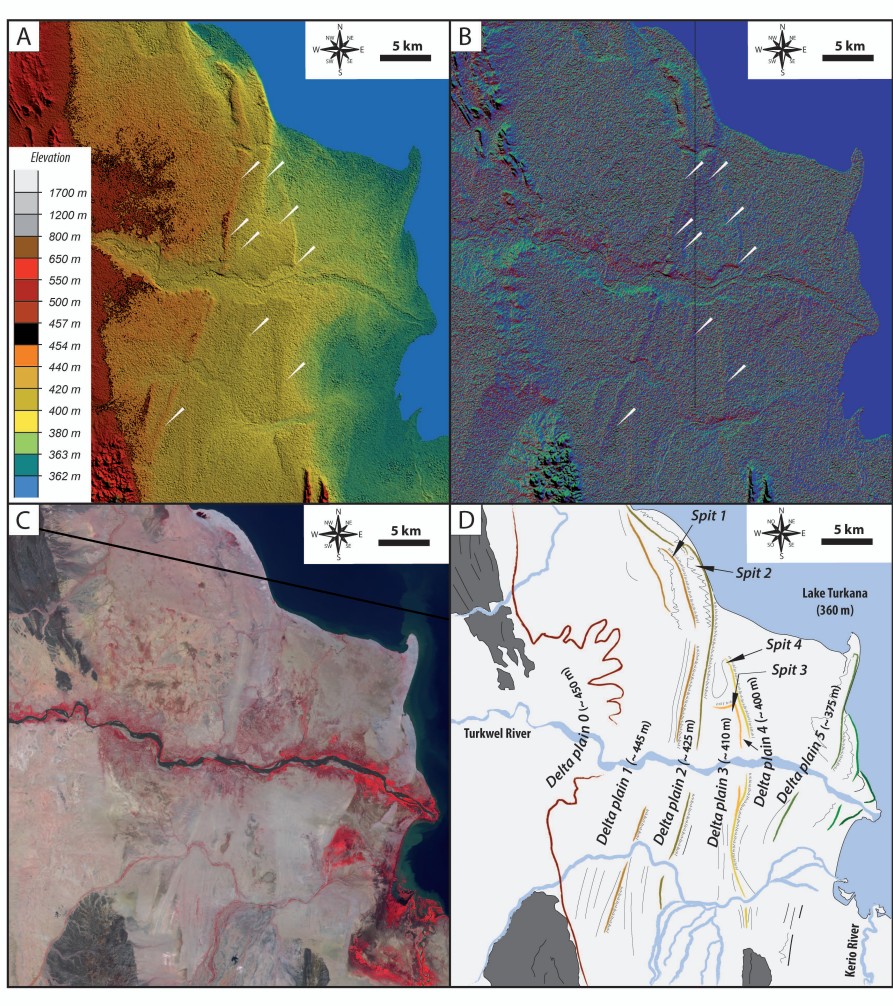











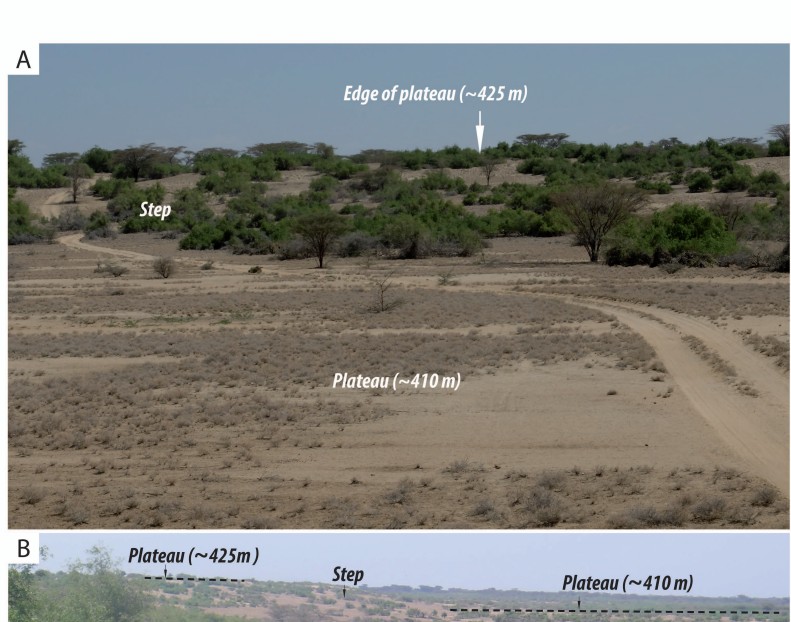



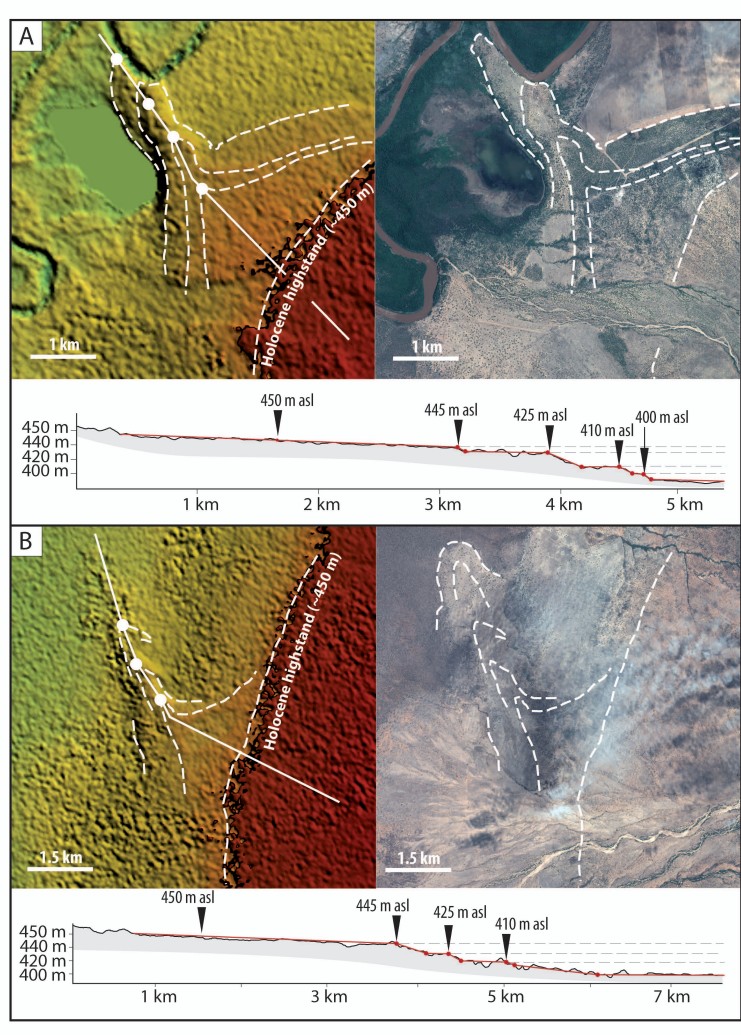





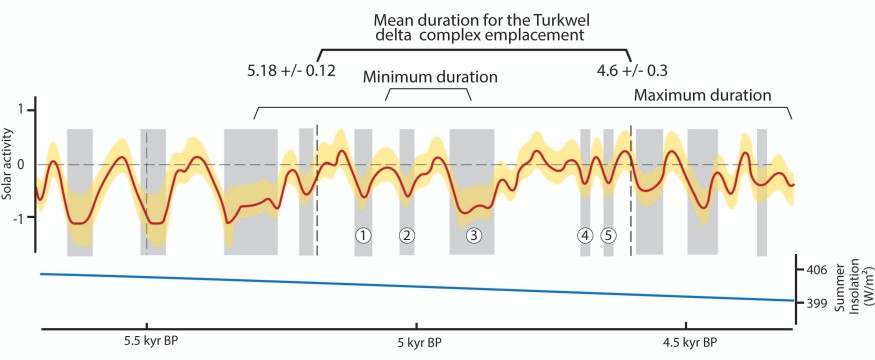