# Peer review of "Alexis Nutz1, Mathieu Schuster1"

_Solid Earth, 2016_

## Referee Comment (RC1) · Anonymous Referee #1 · 12 Sep 2016

GENERAL APPRECIATION

I reviewed two previous versions of this manuscript submitted originally to a different journal, and production of successive improvements to the manuscript has been a learning process on the part of the authors. Their decision to withdraw the submission at the other journal seems to indicate that they had lost confidence in their capacity to accommodate my remaining criticisms (and perhaps also criticisms by the other reviewer). However, the present manuscript is actually close to being acceptable for publication, except for one crucial defect in the presentation of the results that really must be corrected. I hope that my comments below will stimulate the authors to make the requested changes, such that this interesting study can finally see the light of day

in print.

MAJOR COMMENTS

The study described in this paper is an original application of geomorphological data to reconstruct past climate and hydrological change in a semi-arid region of tropical Africa where detailed paleoclimate proxy records are scarce. The geomorphological analysis itself appears technically sound, but the study's significance is undermined by the rather unsatisfying paleohydrological and paleoclimatological interpretation of the data. It is exactly the innovative use of a beach-ridge sequence as paleoclimate proxy (and the way in which it may address a much-debated climate question) that is worth publishing in Solid Earth, but only if the argument linking the temporal spacing of the beach ridges to climate change is sound and robust. As currently presented the authors' central claim that the chronosequence of mid-Holocene beach ridges at Lake Turkana in northern Kenya (reflecting the lake's stepwise regression during a prolonged drying phase at the end of the African Humid Period) can be correlated with variations in solar irradiance remains far out on a limb, creating a risk that the research community specializing in African climate history will dismiss the results of this study.

However, this does not need to be its fate. With relatively minor additional analysis the present submission could improve sufficiently to be considered technically sound. With most issues troubling the original paper now resolved or accommodated, the principal problem undermining the authors' inferences is the lack of good chronology. First they correctly state that the Turkana beach ridge sequence covers the period between 5.18 ± 0.12 and 4.6 ± 0.3 ka BP (lines 84-85), i.e. a period lasting between 160 and 1000 years (lines 89-90). In previous versions of the manuscript they then proceeded to claim a prominent influence of past variation in solar irradiance on the mode of lake regression (and thus mid-Holocene rainfall variability), by noting coincidence between the number of beach steps with the number of known anomalies in solar irradiance between 5180 and 4600 years ago (and simply neglecting dating uncertainty on the start and end points of this time bracket). In reality the beach sequence interval encompasses between two and ten solar irradiance anomalies depending on whether it lasted as little as 160 years or as much as 1000 years. The leap of faith required to accept that one can take the mean duration of the covered period (580 years) to claim a 1-on-1 correspondence between five beach steps and five solar irradiance anomalies is simply too great. In all honesty, this correspondence is highly tentative, and it must be presented as such.

In the present manuscript, Nutz and Schuster try to resolve the issue by indicating the minimum and maximum duration of the chronosequence interval on the solar-activity curve (Fig. 6), and by down-playing the 1-on-1 correspondence between the mean duration of the interval and five solar-activity minima as an 'interesting match' (line 197). But in so doing, they resign to an equal probability of about 11% that the chronosequence interval encompasses exactly five or any number of solar minima between two and ten. Evidently, this does disservice to the data. The real solution is to calculate the probability curve for the age difference between the start and end dates of the chronosequence interval. Probability curves for the latter can be obtained by 1) rerunning the calibration of the 14C date reported by Bloszies et al. (2015) for the start of the interval; and 2) calculating the Gauss curve describing uncertainty on the end date for the interval, based on the regression between OSL and 14C dates presented by Forman et al. (2014). The probability curve for the difference will determine the % chance that the chronosequence interval covers a period of about 600 years (equivalent to 5 solar minima), versus the probabilities that it covers a significantly shorter or longer period (and thus less or more solar minima, respectively). The end result will likely assign a probability of between 30 and 40% to a 1-on-1 (5-on-5) match. This would still warrant a tentative conclusion, but at least the analysis reaching this conclusion would be robust.

MINOR POINTS

Compared to previous versions, the present manuscript has much improved on the presentation of climate dynamics and forcing mechanisms. Comments below include

some suggestions for further improvements in this regard. The Acknowledgement section mentions that the authors have used an editing service to improve the English text, but unfortunately quite a large number of linguistic errors remain (perhaps due to alterations made after the proof reading?), as well as phrasing that is linguistically correct but convoluted or ambiguous.

Title: I recommend modifying to "Stepwise drying of Lake Turkana at the end of the African Humid Period: a forced regression modulated by solar activity variations?"

Abstract: Please compare the version below with the original text, and use it as reference to make textual improvements throughout the paper.

"Although timing of the termination of the early-Holocene African Humid Period (AHP) is now relatively well established, modes and controlling factors are still being debated. Here, through a geomorphological approach, we characterize the evolution of the final regression of Lake Turkana at the end of the African Humid Period. We show that lake-level decline during this period was not continuous but stepwise and consisted of five episodes of rapid lake-level decline separated by episodes of slower rates lake-level decline. Whereas the overall regressive trend can be attributed to decreasing regional precipitation associated with the gradual reduction in northern hemisphere summer insolation controlled by orbital precession, we focus discussion on the origin of the five periods of accelerated lake-level decline. We propose that these are due to temporary reductions in rainfall across the Lake Turkana basin associated with repeated westward displacement of the Congo Air Boundary during minima in solar activity."

Lines 35-37: delete "Depending on the location", and formulate this sentence as follows: "The mid-Holocene termination of the AHP is thought to have been either abrupt (e.g., deMenocal et al., 2000), gradual (e.g., Kröpelin et al., 2008) or time-transgressive depending on location (e.g., Shanahan et al., 2015), an ongoing debate highlighting the variable responses of proxies to dominant forcings and the complex interactions among the multiple components of the local environment."

Lines 41-42: "...the final regressions of African lakes are presented with relatively constant rates of lake-level decline (insert refs. here)".

Line 49: delete "and well developed".

Lines 50-52: "These coastal features represent a valuable paleohydrological archive that contributes to understanding of the evolution of Lake Turkana during the AHP".

Lines 53-54: define 'forced transgression' here at its first mention, instead of on line 121.

Line 66, quantify "multi-metre".

Lines 147-148: "repeated pulses of accelerated [. . .] emptying of a magma chamber..."

Line 163, "half-precessional forcing": to my knowledge, neither of the cited references discusses half-precessional forcing, which is a pattern of orbital forcing only observed near the equator where northern and southern hemisphere monsoon systems interact (see, for example, Verschuren et al. 2009 Nature). Clearly monsoon circulation over Lake Turkana responds to simple precessional forcing (i.e. with c.23k-year periodicity) paced by northern hemisphere summer insolation, not half-precessional forcing (with c.11.5-kyr periodicity).

Lines 187-189, rephrase as "These lakes are considered 'amplifier lakes' (Street-Perrott and Harrison, 1985) for which relatively modest changes in climate are amplified into significant lake-level fluctuation due to their specific catchment morphology." First, amplifier lakes were first defined by Street (1980: The relative importance of climate and local hydrogeological factors in influencing lake-level fluctuations. Palaeoecology of Africa 12,137-158). Recently several publications from the group of Martin Trauth in Potsdam re-emphasized their value as paleohydrological recorders. Crucially not only catchment morphology is important, but also a large catchment-to-lake area ratio so that strong lake-surface-evaporation is compensated with substantial river (and subsurface) inflows. Therefore, amplifier lakes are typical for semi-arid tropical and

subtropical regions. All this deserves a few more lines of explanation here.

Line 194, "major solar activity minima": as noted in my previous review, the amplitude of these claimed 'drastic decreases' in 'insolation' are only between c.0.02 and 0.07% of total solar irradiance. Only very specific regional climate dynamics combined with a very specific lake hydrology (cf. above) can translate this small forcing into a major step-wise lake-level decline. I repeat my plea for a well-structured argument in the paper showing that the authors understand how this connection works exactly.

Lines 211-213, rephrase as "Geomorphic analysis (i.e. trajectory analysis) revealed for the first time a stepwise lake-level decline of Lake Turkana during its final forced regression at the end of the AHP. Five rapid falls in lake level were identified, intercalated with periods of slower lake level fall. We suggest that the. . ."

---

## Author Comment (AC1) · 16 Sep 2016

ANSWERS TO ANONYMOUS REVIEWER 1

We are grateful to reviewer 1 for his review and time he spent on the manuscript. In the following, we present answers to his comments. We first bring some remarks responding to the general consideration of reviewer 1. Second, we discuss the chronology issue that corresponds to the main concern of reviewer 1. Please not that all recommendations are now integrated. Third, we integrated recommendations proposed in the "Minor comments" section.

General appreciation

[Figure]

We would like to precise to reviewer 1 that we decided to withdraw our submission to a previous journal because reviewer 1 was already asking for more precise ages that are not possible to provide for several reasons. The main reason for this is because the accuracy of the dating methods (14C or OSL) does not allow working at such a short time scale of 10 to 100 years. Opposite affirmation would reveal an unfounded (almost foolish) confidence in such methods. Second, during our field trip in the area, we did not find any potential material usable for accurate and meaningful radiocarbon dates (we mean especially charcoals because mollusk shells would introduce more uncertainties considering the unknown reservoir effect). In addition, sampling for OSL would need important logistics in order to remove large amount of eolian deposits before reaching the material to date. This is not relevant especially for a method that does not provide the requested precision. Finally, we hope that reviewer 1 will consider those limits in his potential new review (please find our answer in the "Major comments" section below). Moreover, we would like to remind that the core of the paper is not to refine the age or the timing of the termination of the AHP in Lake Turkana, but rather to understand the drying trend, that up to now was considered as relatively linear in Lake Turkana as in other lakes of Africa. In our opinion, this information is essential regarding our necessity to understand how lakes respond to transition from wet to dry period in this part of the world. Nevertheless, we are grateful that reviewer 1 who acknowledges our learning as well as successive improvements of this paper as stated in its general appreciation "I reviewed two previous versions of this manuscript submitted originally to a different journal, and production of successive improvements to the manuscript has been a learning process on the part of the authors." Thanks again. We are now convinced that the presently submitted version of this paper to SE is the best ever.

Major comments

The major comment of reviewer 1 is "the lack of good chronology". For this paper, we rely on age-models that are published in international peer review journals. Since the aim of this paper is not to refine this age model (which appears difficult to do if one considers the precision that can be reached by applicable methods), we did not focused our work on providing new ages which would have been at best in the same precision range as the already published ages. Reviewer 1 continues to think that more precise ages can be given for this study. In our opinion, significant more precise ages are not possible to get and the diverse calculations proposed by reviewer 1 are just pseudo-quantitative gestures, which first only propagate more and more errors and second complicate the manuscript. Worth noting, we however decided to follow the recommendation of reviewer 1 because it does not change something to the core of the manuscript. First, we rerun calibration (curve INTCAL13) of the 14C age of the beginning of the final forced regression. This age is from the sample SNU12-589 (Bloszies et al., 2015). A new calibration provides an age of $5.14 \pm 0.18$ ka cal BP ($\sigma 2$). This is very close that date given by Bloszies et al. (2015) used in the previously submitted manuscript (fortunately it is). Concerning the age of termination of the final regression, we proposed to consider the age of $4.58 \pm 0.25$ yr BP (sample OSL23/1.30; Forman et al., 2014). This age is an OSL age. However, to follow up recommendation of reviewer 1, we converted this OSL age into a radiocarbon age. Based on 6 examples for which OSL and radiocarbon ages exist, we carried out statistic correlation between OSL ages and their radiocarbon equivalent ages (data Fig. 4; Forman et al., 2014). We obtained a correlation function (age(OSL)=0.98386063*age(14C(calibatred)); b( the intercept) has been forced to 0). This correlation gives an equivalent radiocarbon age of $4.65 \pm 0.3$ ka cal BP ($4.13 \pm 0.24$ ka 14C BP) for the end of the final regression. Once again, this age is very close to the age proposed in the submitted manuscript. Finally, we considered the maximum potential time interval during which the final regression took place (4.57 to 3.90 ka 14C BP) and assigned it a mean age ($4.23 \pm 0.34$ ka 14C BP). After calibration of this mean age, the probability curve suggests that there is a 44% of probability that the regression precisely occurred between $5.14 \pm 0.18$ and $4.65 \pm 0.3$ ka cal BP. This is slightly better than the 30-40% estimated by reviewer 1. This is now stated in the manuscript as recommended by reviewer 1 and the "Chronological framework" section has been extended to explain such processing. However, once again we would like to remind that this paper provides firm evidences for a stepwise regression at the end of the AHP. In our knowledge, this is the first report of such dynamics during the final regression of the AHP. These are the facts that anyone can observe and this is the core part of the paper. We then only discuss a potential forcing, subsequently proposing a mechanism. We never argue that the role of this forcing and the veracity of the mechanism are firmly established. We ask to readers that the discussion be considered as a discussion and we are looking forward for alternative explanations based on other forcings for such a decadal to centennial repeated/cyclic lake level evolution that can impacts the general architecture of a delta.

Minor comments

Authors are grateful to reviewer 1 for his suggestions that improve the text. All minor comments proposed by reviewer 1 are now integrated.

Best regards Alexis Nutz and Mathieu Schuster

Please also note the supplement to this comment:
http://www.solid-earth-discuss.net/se-2016-95/se-2016-95-AC1-supplement.pdf

**Supplement:**

**Stepwise drying of Lake Turkana at the end of the African Humid Period:**
**a forced regression modulated by solar activity variations?**

**Alexis Nutz[1], Mathieu Schuster[1]**

*Institut de Physique du Globe de Strasbourg (IPGS), UMR 7516, Centre National de la*

*Recherche Scientifique, Université de Strasbourg, École et Observatoire des Sciences de la*

*Terre, 1 rue Blessig, 67084 Strasbourg, France*

*\*Corresponding author* (nutz@unistra.fr)

**Running head:** Lake Turkana drying at the end of the AHP

**Keywords:** East African Rift System; Turkwel delta; Lake-level; Holocene; Solar activity

**Abstract**

Although timing of the termination of the African Humid Period (AHP) is now relatively well-established, modes and controlling factors are still being debated. Here, through a geomorphological approach, we characterize the evolution of the final regression of Lake

Turkana at the end of the AHP. We show that lake level fall during this period was not continuous but stepwise and consisted of five episodes of rapid lake-level fall separated by episodes of slower rates of lake-level fall. Whereas the overall regressive trend can be attributed to decreasing regional precipitations due to the gradual reduction in northern hemisphere summer insolation controlled by orbital precession, we focus discussion on the origin of the five periods of accelerated lake-level fall. We propose that these are due to temporary reductions in rainfall across Lake Turkana area associated with repeated westward displacement of the Congo Air Boundary (CAB) during minima in solar activity.

**1. Introduction**

The African Humid Period (AHP), *c.* 14.8 to 5.5 ka cal BP, is a major climate period that was paced by orbital parameters (i.e. precession) (deMenocal et al., 2000; deMenocal and Tierney, 2012; Bard, 2013; Shanahan et al., 2015) and that markedly impacted environment, ecosystems, and human occupation of Africa over several millennia (Bard, 2013). An increase in rainfall during this climate period led to the rise and highstand of numerous African lakes (Street and Grove, 1976; Tierney et al., 2011). The end of the AHP was characterized by the establishment of more arid conditions, leading to dramatic lake level falls (Street-Perrott and Roberts, 1983; Kutzbach and Street-Perrott, 1985). This aridification forced Neolithic populations to adapt to more limited resources (Kuper and Kröpelin, 2006) and represents a recent example of major climate change. The mid-Holocene termination of the AHP is thought to have been either abrupt (deMenocal et al., 2000), gradual (Kröpelin et al., 2008) or time-transgressive (Shanahan et al., 2015) depending on location, an ongoing debate highlighting the variable responses of proxies to dominant forcings and the complex interactions among the multiple components of the local environment (e.g., deMenocal, 2000; Renssen et al., 2006; Liu et al., 2007; Tierney and deMenocal, 2013; Shanahan et al., 2015). However, drying trends remains poorly-constrained and as a consequence the final regressions of African lakes are presented at relative constant rate of lake level fall (e.g., Garcin et al., 2012; Forman et al., 2014; Morrissey and Scholz, 2014; Junginger et al., 2014; Bloszies et al., 2015). In this study, we investigate the drying trend of Lake Turkana and evidence for the first time that the final regression was not continuous through time revealing a more complex process than previously envisaged. Thus, understanding the mode of African lake regressions appears as particularly relevant in the context of projecting future global climate change impacts on the African continent (e.g., Patricola and Hook, 2011), especially in term of evolution of water resources from large lakes.

Lake Turkana is one of the great lakes of the East African Rift. It is considered as a

Wind-driven Waterbody (Nutz et al., in press) that developed abundant wave-dominated coastal features all along its shoreline. These coastal features represent a valuable paleohydrological archives that contributes to the understanding of the evolution of Lake

Turkana during the AHP (Garcin et al., 2012, Forman et al., 2014; Bloszies et al., 2015).

However, the detailed and continuous evolution of lake level during the final forced regression (i.e., basinward migration of the shoreline associated with a base-level fall)

marking the end of the AHP has not been already documented. Here, the delta complex of the

Turkwel River (Fig. 1) that developed during the final forced regression of Lake Turkana is examined using trajectory analysis (Helland-Hansen and Hampson, 2009). Finally, we highlight variations in the rate of lake level fall during the regression. We interpret those variations as markers reflecting variable rate of decrease in precipitation during the crucial period corresponding to the terminal phase of the AHP. Subsequently, we discuss potential forcings responsible for the regressive pattern of Lake Turkana with a primary focus on the role of the Sun and short-term variability of insolation.

**2. Materials and methods**

The data set is comprised of satellite imagery and a digital elevation model (DEM). A

recently obtained SRTM1 (Shuttle Radar Topography Mission (Rabus et al., 2003)) is available for the entire Lake Turkana area. This DEM is produced by radar interferometry with a one arcsec (approximately 30 m) horizontal grid spacing and an approximately 5 m absolute vertical error (Rosen et al., 2001; Tighe and Chamberlain, 2009). In addition, high- resolution (<1 m) PLEIADES and (5 m) SPOT 5 images (©CNES 2012, Airbus DS/ SPOT

Image) were used to focus on selected areas. This data set was processed using GIS software (Global Mapper 15 software; Blue Marble Geographics, Hallowell, ME, USA) to provide a high-resolution 3D view of the geomorphological features. Topographic profiles, elevation differences, and slope values used for the trajectory analyses were obtained using Global

Mapper 15 software.

**3. Chronological framework**

Humid conditions related to the AHP broadly prevailed over Africa from 14.8 to 5.5 ka cal

BP (deMenocal et al., 2000; Shanahan et al., 2015). Several lake level curves associated with

Lake Turkana evolution provide records of the regional moisture history over the Holocene (Garcin et al., 2012; Forman et al., 2014; Bloszies et al., 2015). Based on surveys of raised

Holocene beach ridges coupled with dated archeological sites, these studies provide a relatively robust chronological framework for the final regression at the end of the AHP.

Garcin et al. (2012) initially estimated the onset of the final lake level fall in Lake Turkana at

*c*. 5.27 ± 0.36 ka cal BP. Subsequently, Forman et al. (2014) proposed that the age of this final regression occurred between 5.5/5.0 to 4.6 ka cal BP associated with a lake level change from 440 to 380 m asl. Finally, Bloszies et al. (2015) proposed an onset of the final regression of the AHP starting at 5.18 ± 0.12 ka cal BP (dating of a shell at 90 m above the modern Lake

Turkana) and finishing at 4.58 ± 0.25 ka BP (OSL age reused from Forman et al., 2014; sample UIC2319) associated with a lake level grading from 450 to 375 m asl. Based on these published data, we carried out minor complementary processing in order to refine the chronology. First, we recalibrated sample (SNU12-589) considered to provide the age of the onset of the final regression. Using a most recent curve (INTCAL13; Reimer et al., 2013), the onset of the final regression is now given at 5.14 ± 0.18 ka cal BP (4.51 ± 0.06 ka $^{14}$C BP).

Second, we converted the OSL age (4.58 ± 0.25 ka BP; sample OSL23/1.30) that is considered to represent the end of the final regression (Forman et al., 2014) in radiocarbon age. Indeed,

Forman et al. (2014) provided 6 samples that were dated by both OSL and radiocarbon methods. Even the limited number of samples, we then processed a linear regression in order to propose a statistic relationship between OSL and radiocarbon ages. At the end, based on this correlation ($age_{(OSL)}=0.98386063*age_{(^{14}C(calibatred))}$; b( the intercept) has been forced to 0;

$r^2=0.9942$), the age of the end of the final regression is now estimated at 4.65 ± 0.3 ka cal BP

(4.14 ± 0.24 ka $^{14}$C BP). As such, based on thise most recent available age-model, the final regression of Lake Turkana at the end of the AHP would, at the longest, span a period from

5.32 to 4.35 ka cal BP. At a minimum, the final regression would have occurred between 4.96

and 4.95 ka cal BP. This implies a duration ranging between 10 to 1030 years, with a mean duration of 510 years for water level to decrease from the Holocene highstand (450 m asl) to the lowstand (375 m asl). Considering the largest potential time interval during which the final regression occurred (i.e., interval between 4.57 and 3.90 ka $^{14}$C BP ), a mean age of 4.23

± 0.33 ka $^{14}$C BP is established in order to allow a calibration and then to provide a probability curve. At the end, calibration reveals a 44% of probability that the final regression precisely occurred between 5.14 ± 0.18 and 4.65 ± 0.3 ka cal BP. 
[revised manuscript text omitted]
., 2013). Especially in Lake Turkana, the potential expression of the 11-year sunspot cycle has already been deciphered through time-series analysis for sediments associated with the last 4 ka (Halfman et al., 1994). Within some African lakes, several authors link more arid periods with solar activity minima (Stager et al., 2002 and Junginger et al., 2014) and Lake

Turkana is one of them. The capacity of those lakes to record changes in paleohydrology attributed to variations in solar activity may rely to the fact that these lakes are very sensitive.

Indeed, they are considered as "amplifier lakes" (Street-Perrott and Harrison, 1985) for which relatively modest changes in climate are amplified into significant lake-level fluctuation due to their specific morphology. As an amplifier lake, Lake Turkana could be more sensitive to precipitation changes from small variations in insolation as those generated by modifications in solar activity.

Coupling the proposed chronological framework with the solar activity curve from

[revised manuscript text omitted]

---

## Referee Comment (RC2) · Anonymous Referee #2 · 21 Sep 2016

Review of Nutz and Schuster: Stepwise drying of Lake Tana at the end of the African Humid Period

General Comments This is an interesting paper which provides new geomorphic evidence from Lake Turkana, northern Kenya on the termination of the African Humid Period c. 5 ka BP. The analysis of shoreline data presented here is worthy of publication, adding new insights into the nature of shoreline regression and progressive drying during the termination of the AHP. I do however have concerns about the lack of chronological control of these features in relation to the claims which are made regarding solar forcing. The manuscript is generally well presented, although there are typographical and grammatical errors. Substantive comments and suggested editorial

corrections which require attention are outlined below.

Specific Comments Section 3, Chronological framework: I appreciate that the authors give references to the original studies which provide the chronological framework but the techniques which generated these ages should at least be referred to in this section (e.g. line 84 specify this is a radiocarbon date). The authors have not taken into account that the errors quoted for the Forman et al date of 4.6 +/- 0.3 ka BP are 1 sigma (see their Table 7, note e). So, the minimum age of the end of the lake level decline could be 4.0 ka BP, extending the maximum possible duration of the phase of lake level decline. If 2 sigma ages for the radiocarbon dates are applied in the analysis here, the same should be applied to OSL ages. Trajectory analysis: Could the authors please explain why 'fall 2' identified by trajectory analysis is only present in 2 of the 3 transects (not clear from lines 103-105)? Furthermore, in one of these, P2, the gradient of the slope is not very distinct (only 0.3°) compared with other falls which are identified by this method.

Solar forcing of stepwise regression: The case for the link between stepwise regression and solar minima is overstated given the lack of age control for these features. I appreciate that it is not possible for the authors to resolve given the lack of material available in such settings and errors associated with chronological methods which could be applied. However, it is stretching the evidence to far to correlate five stepwise regressions with five periods of solar minima coinciding with the calculated mean of the duration of lake level decline (and these figures need to be revised to account for 2 sigma errors on OSL ages). There is no justification for the statement made in lines 176-178. It is fine to pursue this as a possible line of investigation, but there are other factors to considered such as multidecadal climate variability (e.g. AMO?). Feedbacks such as the impact of changing vegetation cover and evapotranspiration on water balance may also contribute. The final paragraph of the discussion needs more supporting references for the mechanistic link that is postulated between solar minima / maxima and shifts in the shifts in the Congo Air Boundary, at present this is not convincing. Section

5.2 overall needs a much clearer statement that the chronological uncertainties mean that the link to solar forcing must be considered as tentative.

Minor / Editorial Comments There are a number of typographical errors and some phrases which could be constructed more clearly: 1. Suggest altering the 'running head' to 'Drying of Lake Turkana at the end of the AHP' 2. Line 17: should read 'rather stepwise, consisting of' 3. Rephrase sentence beginning on Line 18 of the abstract, 'Even the overall regressive trend. . .' – the first part does not make sense and two separate sentences would be better. 4. Line 28: should read 'impacted on the' 5. Line 35: I wouldn't use the word 'recent' in this context. 6. Line 40: should read 'drying trends remain' 7. Line 43: I don't think others have suggested that lake level fall was constant / continuous but have provided chronological constraints on beginning and end of the regression. Several authors explicitly state that the end of the African Humid Period was marked by fluctuating conditions. It would be useful to add in brackets the approximate age of the final regression that you are referring to for clarity. 8. Line 44: replace 'appears as' with 'is' 9. Line 46: should read 'terms of' 10. Line 52: it would be appropriate here to acknowledge some of the earlier studies on lake levels / shorelines at Lake Tana (e.g. Owen et al, 1982; Butzer 1971/1980). 11. Line 54: replace 'already' with something like 'clearly' 12. Lines 57: sentence starting 'Finally' should not come before sentence starting 'subsequently' 13. Line 78: 'archaeological' (depending on journal house style) 14. Line 145: replace 'volcanism event' with 'although volcanic activity' and replace 'even' on the following line with a comma. 15. Line 160: reference needed at end of sentence. Also rephrase to make clear that the AHP began before the early Holocene. 16. Line 162: should read 'reduction in' 17. Line 169: should read 'periods of higher insolation' – could specify summer here. 18. Line 193: remove 'the' from before Lake Tana 19. Line 211: should read 'stepwise' 20. Line 218: should read 'plateau' and start sentence with 'The'